# Study on the Fingerprint and Atmospheric Activity of Volatile Organic Compounds from Typical Industrial Emissions

**DOI:** 10.3390/ijerph20043517

**Published:** 2023-02-16

**Authors:** Xin Gu, Kaitao Chen, Min Cai, Zhongyi Yin, Xingang Liu, Xingru Li

**Affiliations:** 1Department of Chemistry, Analytical and Testing Center, Capital Normal University, Beijing 100048, China; 2College of Resource Environment and Tourism, Capital Normal University, Beijing 100048, China; 3State Key Laboratory of Water Environment Simulation, School of Environment, Beijing Normal University, Beijing 100875, China

**Keywords:** VOCs, source profiles, ozone formation potential (OFP), secondary organic aerosol potential (SOA), emissions

## Abstract

China is prone to severe surface ozone pollution in summer, so it is very important to understand the source of volatile organic compounds (VOCs) to control ozone formation. In this work, the emission characteristics of 91 VOC components from the plastic products industry, packaging and printing industries, printing ink industry, furniture manufacturing and vehicle manufacturing industries were studied. The results show that there are significant differences between these sources, and for the plastic products industry, alkanes (48%) are the most abundant VOCs. The main emission species in the packaging and printing industry are OVOCs (36%) and alkanes (34%). The proportion of OVOCs in the printing ink (73%) and furniture manufacturing industries (49%) is dominated by VOC emissions; aromatic hydrocarbons (33%), alkanes (33%), and OVOCs (17%) are the main emission species in the vehicle manufacturing industry. At the same time, the ozone generation potential (OFP) and secondary organic aerosol formation potential (SOA) of anthropogenic VOC emissions were evaluated, and the top 10 contributors to OFP and SOA were identified. Toluene, o-xylene, and m-xylene had a significant tendency to form OFP or SOA. Then, a health risk assessment of VOC components was carried out. These data can supplement the existing VOC emission characteristics of anthropogenic emissions, thus enriching the research progress of VOC emission sources.

## 1. Introduction

Volatile organic compounds (VOCs), as key precursors of atmospheric ozone and secondary organic aerosol pollution, are attracting increasing global attention in terms of their emission sources and environmental impacts [1,2,3]. Additionally, VOCs contain a variety of cancerogenic substances, such as benzene, toluene, ethylbenzene, xylene (BTEX), and aldehydes [4,5,6], leading to a series of negative impacts on human health. VOCs generally include C2–C12 nonmethane hydrocarbons, oxygenated volatile organic compounds (OVOCs), halohydrocarbons and other organic matter [7,8]. They are mainly generated by natural and anthropogenic sources, such as the combustion of fuels and wood and from industrial emissions [9,10,11]. Industrial emissions are considered the largest anthropogenic source of VOCs in China [12,13]. Relevant research has shown that VOCs emitted by industrial sources increased from 15.3 Tg in 2011 to 29.4 Tg in 2013 at an annual average growth rate of 38.3% [14].

Due to large differences in raw and auxiliary materials, technological processes, VOC emission links, and VOC post-treatment, the source and emission characteristics of industrial VOCs are very complicated. An in-depth study of VOC composition characteristics in various industries is the key to revealing the formation mechanism of ozone pollution, as well as secondary organic aerosols in PM_2.5_. Consequently, the source spectrum of VOC emissions from industries has been studied in many cities in China [10,13]. Zhong and Zhang et al. [15,16] established source component spectra for automobile manufacturing, shipbuilding, wood painting, metal surface painting, and the petrochemical industry in the Pearl River Delta region to obtain VOC emission concentrations and assess their atmospheric reactivity. Mo et al. [17] explored the VOC emission characteristics of ship container, ship building, wood, and auto painting industries and found that aromatic compounds, such as toluene, xylene, and ethylbenzene, accounted for 79–99% of total VOCs. Wang et al. researched the source profile and chemical reactivity of VOCs emitted by solvent use in Shanghai and found that emissions from imported indoor solvents, ~50% of the total mass concentration, were contributed to by aromatics and ~30% were contributed by alkanes [18]. Wu et al. analyzed the sources of anthropogenic volatile organic compounds in typical solvents and industries [19,20]. The results showed that the average value of TVOCs in eight hours of six types of furniture panels ranged from 0.1 to 0.7 mg/m^3^. Analysis of the VOC composition spectrum shows that the main emission species are toluene, formaldehyde, n-nonane, methanol, propane, ethane, etc. Although there are many studies about VOC source profiles in China, some significant differences in source compositions have been reported between different regions. These earlier domestic VOC source profiles are limited; however, they are mainly concentrated in economically developed regions, such as the Pearl River Delta (PRD), Yangtze River Delta (YRD), Chengdu–Chongqing region, and Beijing–Tianjin–Hebei region (BTH) [21]. Due to numerous industrial categories with complex production processes, the source and emission characteristics of industrial VOCs are very complicated. Furniture manufacturing, equipment manufacturing, automobile manufacturing, paint coating, and pharmaceutical industries are all considerable sources of VOCs, but limited studies have been conducted on these industries in Yuncheng. This paper focuses on the Yuncheng area. Due to the terrain characteristics and industrial structure of the Yuncheng area, air pollution is relatively serious. Complex atmospheric pollution such as photochemical haze in summer is widespread, but research on Yuncheng’s composition spectrum of industrial sources is still relatively lacking. Therefore, this study collected typical artificial source samples for analysis. To obtain the emission characteristics of VOC sources and explore the differences between different sources, the relevant impacts of ozone formation potential (OFP), source reactivity (SR) and secondary organic aerosol potential (SOA) on the atmospheric environment were analyzed and the VOC emission inventory of typical solvents were established using industries in Yuncheng city. To provide supporting data for Yuncheng VOC key emission industry priority control, a “one enterprise, one policy” plan was developed.

## 2. Materials and Methods

### 2.1. Industry Source Identification

According to the results of the source analysis of VOCs in Yuncheng city in Shanxi Province [22], anthropogenic emissions accounted for 92% of the total VOCs, of which 40% were contributed by solvent use and industrial emissions. In this study, the plastic products industry, packaging and printing industry, printing ink industry, furniture manufacturing, and vehicle manufacturing industry were selected as the research objects, and the source emission characteristics of typical anthropogenic emission sources in Yuncheng city were studied. The sampling points are shown in Figure 1. Yuncheng is an industrialized area, and the sampling point selected in this study is located near the downtown area of Yuncheng. In this study, companies with relatively large scale and complete production lines in the local area are equipped with waste gas treatment facilities in the main process links involved in VOCs production.

### 2.2. Sample Collection

The VOC sampling method [23,24] is based on the “HJ732-2014 Air Bag Method for Sampling Volatile Organic Compounds from Exhaust Gas of Fixed Pollution Sources” and other published results. Organized sampling was conducted at the VOC exhaust outlet using a bong, and the flow rate was controlled by a flow valve. The sampling time was approximately 5 min. The atmospheric sampling pump was used to collect the VOC workshop or factory at 100 mL/min. The sampling height was approximately 1.5 m, and the sampling time was approximately 20 min. Before sampling, an airbag wash was carried out with ambient air. This survey industry was equipped with VOC exhaust gas treatment devices before and after emission treatment for sample collection. After sampling, the samples were coded and sent to the laboratory for component analysis.

### 2.3. Sample Analysis Method and Quality Control

The quantitative method of VOCs refers to the TO-14 and TO-15 analysis methods of the US Environmental Protection Agency (EPA). The pre-concentration system uses the Entech7200 concentrator to concentrate the gas sample in three steps to remove H_2_O, CO_2_, N_2_ and O_2_ gas except for VOCs in the gas. Driven by high-purity helium gas (≥99.9%), the concentrated VOC components are desorbed into the capillary column of the GC and entered into the gas chromatography–mass spectrometry (GC–MS/FID, trace DSQII, Thermo) system for separation and detection. The initial temperature of the chromatographic column was kept at 30 °C for 10 min, raised to 90 °C at 3.8 °C/min for 1 min, raised to 130 °C at 15 °C/min, raised to 200 °C at 8.5 °C/min, and held for 10 min. High-purity helium (≥99.9%) was further purified by filtration and deoxygenation, and the sample was injected in constant pressure mode with a pre-column pressure of 137 kPa; the transfer line temperature was 200 °C. Mass spectrometry conditions: the ion source temperature was 220 °C; the ionization method was electron bombardment (EI), the ionization energy was 70 eV, and the electron multiplier voltage was 1750 V. The scanning mode was full scanning, the scanning range was 45~200 amu, and the scanning frequency was 4.21 scans/s.

To ensure the stability of the instrument, seven volume fractions of standard compounds were selected to draw the working curve (2 × 10^−9^, 4 × 10^−9^, 6 × 10^−9^, 8 × 10^−9^, 10 × 10^−9^, 20 × 10^−9^, and 40 × 10^−9^). Each concentration was injected three times, the RSD of each concentration was less than 30%, and the average retention time and average response value were taken to establish a standard curve. The detection limit of this method was 0.5 × 10^−9^. The intermediate concentration of the standard curve was used to perform daily calibrations to judge the validity of the working curve, and the quantitative result was within 30% of the actual concentration. Blank analysis was performed once a week. Blank analysis was required to contain no target compounds. The correlation coefficients of the standard curves were all greater than 99%, and the quality control indicators all met the requirements.

### 2.4. Ozone Generation Potential Calculation (OFP)

OFP has been widely used to quantify and predict the photochemical ozone-forming reactivity of VOCs [25], evaluating the reactivity of VOCs based on the estimation of MIR values of VOC species. Since individual VOCs react at different rates and by different mechanisms [26], according to this method, the largest contribution of VOC components in the atmosphere to ozone generation can be quantified, and the relative overall activity of VOCs can be expressed to determine ozone formation key sources and species. The calculation is as follows:(1)$${OFP}_{i}=MIR\times {\left[VOCs\right]}_{i}$$

MIR is the maximum incremental reactivity of VOCs, in g·g^−1^, and its value adopts the research results of Carter et al. [27,28]; VOCs are the mass concentration of the VOC components, in mg·m^−3^. OFP represents the maximum ozone generation potential of this species, and the unit is mg·m^−3^.

### 2.5. Source Activity Calculation (SR)

In this study, the MIR was used to estimate the source reactivity (SR) size. Since most urban areas are in the VOC control zone for ozone generation [29], MIR has strong applicability in VOC control areas. The SR value can be used to compare and analyze the ozone formation potential of the unit emission of VOCs from pollution sources without considering the emission intensity of VOCs, which is suitable for estimating the formation of ozone in large areas [30,31,32]. It has important reference significance for the measurement of ozone generation per unit mass of VOC emissions. It is calculated as follows:(2)$$SR=\sum_{n}^{i}x_{i}\cdot {MIR}_{i}$$

In the formula, SR is the ozone generation coefficient of VOCs, g·g^−1^; “x” is the mass fraction of a single VOC in the total VOC mass; and MIR is the ozone generation coefficient of the VOC component in the maximum ozone increment reaction, g·g^−1^. The MIR values refer to the results of Carter et al. [27,28].

### 2.6. Secondary Organic Aerosol Estimation (SOA)

Aerosol generation potential is a method to judge the generation of organic aerosols. The aerosol generation coefficient (FAC) method was used to estimate the aerosol generation potential of VOCs based on the data from Grosjean’s smoke box experiment. In this paper, this method was used to estimate the amount of SOAs generated by VOC consumption per unit mass. The aerosol generation coefficients of major VOC species were obtained based on data from Grojean’s smoke chamber experiment [33]. The calculation is as follows:(3)$${SOA}_{i}=FAC\times {\left[VOCs\right]}_{i}$$

SOA is the potential of VOC generation of species, and the unit is g·g^−1^; VOCs is the mass fraction of VOC species emitted in the atmosphere, %. The value of FAC is the formation coefficient, %, and refers to the research results of Grosjean et al.

### 2.7. Health Risk Assessment

Health risk assessment is a way to quantitatively describe the relationship between human exposure doses and adverse health effects. Exposure to air pollutants through inhalation, ingestion, and skin contact is generally considered to be the main route of VOC exposure for workers on the shop floor and people living in surrounding areas [34]. Probabilistic estimates of health risks are explanatory tools to reveal the direct or indirect effects of measured pollution levels on human health [35]. Therefore, the hazard index (HI) was estimated according to the US EPA method to assess the hazard risk of inhaling VOCs for the population, and the calculation is as follows:(4)$$HI=\frac{\left(C_{i}\times ET\times EF\times ED\right)}{365\times {AT}_{nca}\times 24}\times \frac{1}{RfC}$$

C_i_: Component concentration of VOCs, mg·m^−3^;

ET: Exposure time, the value is 8 h/d;

EF: Exposure frequency, the value is 250 d/a;

ED: Exposure duration, 30 a;

ATnca: meantime for non-carcinogenicity, 25a;

RfC: for the reference concentration, mg·m^−3^.

## 3. Results and Discussion

### 3.1. VOC Emission Concentration and Composition Characteristics

A total of 91 VOCs were analyzed in this study, including 27 alkanes, 13 alkenes, 1 alkyne, 22 halogenated hydrocarbons, 16 aromatic hydrocarbons, and 12 OVOCs. The VOC emission concentration results and sampling information of various industries are shown in Table 1. The VOC emission concentration of plastic products was 2.97~11.38 mg·m^−3^, packaging and printing was 3.48~50.91 mg·m^−3^, and printing ink was 2.88~43.49 mg·m^−3^. Furniture manufacturing was 1.08~2.25 mg·m^−3^, and vehicle manufacturing was 0.88~12.35 mg·m^−3^. Raw materials, production conditions, treatment processes, and emission types used in different industries led to significant differences in VOC emission concentrations in different production links of various industries. Organized emissions in different industries were equipped with terminal recovery and treatment devices, and the concentration of VOC exhaust gas emitted was significantly reduced after treatment.

At the same time, the composition characteristics of VOC emissions in different processes of the plastic products, packaging and printing, printing ink, furniture manufacturing, and vehicle manufacturing industries were also obtained. Figure 2a shows the proportion of the average concentration species in different industries, and Figure 2b shows the proportion of the concentration value of VOC species in different industries. The emission ratio of components greater than 2% is shown in Figure 2a. There were significant emission differences between the different industries and different processes within the same industry, and OVOCs, aromatic hydrocarbons and alkanes were the main VOC pollution emission sources.

For the plastic products industry, VOCs were mainly generated in the process of batching, molding, mechanical processing, bonding, etc. The process of heating and remodeling is consistent with the research of Zheng et al. [36], which produce a rich variety of alkanes. After plasma + activated carbon adsorption treatment, alkanes were the most abundant VOCs, accounting for 48% of the volatile organic compounds of plastic products, followed by aromatic hydrocarbons (20%), olefins (17%), OVOCs (10%), and halogenated hydrocarbons (5%). The main emission groups were octane, hexene, propanaldehyde, and toluene.

For the packaging and printing industry, the company’s production links involving VOC emissions were concentrated in one area. The production workshop is independent and closed, and the gas generated in the production workshop is first transported through the gas collection device and then processed. The main emitting species were OVOCs (36%) and alkanes (34%), produced by the printing and dyeing process and solvent-based raw materials used in industry. As seen from the figure, the proportion of OVOCs was 65% before the sewage chimney treatment, and the proportion of OVOCs was reduced to 11% after photooxidative plasma + activated carbon adsorption. N-hexal was significantly adsorbed, and the concentration of OVOCs was effectively controlled, among which n-hexal, 1-pentene, pentane, methylene chloride and propanaldehyde were the most abundant species.

For the printing ink industry, OVOCs dominated VOC emissions with a 73% weight percentage, and the main emission groups were ethyl acetate, butenone, ethanol, methylene chloride, and toluene. The high concentration of OVOC emissions may be due to the use of alcohol-soluble inks [12]. Ethanol is commonly used as a water-based ink diluent, ethyl acetate mainly exists in solvent-based inks, and methylene chloride is widely used as a curing agent. The main emission components of VOCs were similar to the typical types of coatings and thinners used as raw materials, indicating that the emission characteristics of the ink industry are highly correlated with raw materials.

In the furniture manufacturing industry, the emissions were still OVOCs (49%), and ethyl acetate in different process links, propane, ethanol, toluene, and ketone of pentane and butene. Emissions are the main components of toluene, and ethyl acetate groups are divided into the typical composition of paint and thinner. Chimneys and the kinds of unorganized workshop VOC emissions are similar, but their proportion is different. It can also show that the process of glue and paint mixing in the furniture manufacturing industry is greatly affected by the raw materials.

For the whole vehicle manufacturing industry, the main emission species were aromatic hydrocarbons (33%), alkanes (33%) and OVOCs (17%). In automobile manufacturing, the main emission species are alkanes after the combustion treatment of electrophoresis, middle coating and finishing paint drying exhaust, gas exhaust cylinders, and the high concentration components, including propane, ethanol, acetone, isopentane and pentane. The VOCs emitted from the gas exhaust cylinder of spray painting and repainting are mainly aromatic hydrocarbons and OVOCs, closer to the components of the coatings used. The high concentration components include butanone, toluene, methylene chloride, ethanol, dichloroethane and ethyl acetate.

### 3.2. VOC Industry Characteristic Emission Components

To further understand the similarities and differences in VOC component emissions in different industries, the VOC components in the five industries investigated in this study were screened, and the species components whose maximum proportion of component mass concentration was greater than 10% of the VOC emissions in this link were selected for characteristic component analysis. It can be seen from Figure 3 that there were obvious differences among different links. The main VOCs emitted by the plastic products industry and the packaging printing industry were pentane, isopentane, 3-methyl-1-butene and propional, etc. The average emission ratio was 10.75%, 13.71%, 7.01%, 6.42%, and 10.10%, 13.38%, 6.54%, 6.01%, respectively. Toluene (4.25%), ethylbenzene (3.19%) and o-xylene (4.50%) were also significantly discharged from the plastic products industry. Dichloromethane (7.66%) and toluene (6.82%) were also major components in the packaging and printing industry. Ethyl acetate (46.05%) and toluene (15.95%) were the most emitted components in the ink industry. In particular, the emission of ethyl acetate was more than 70% in the sample production workshop, which was a typical emission component in the ink industry. Ethanol (20.94%), ethyl acetate (19.49%), propane (12.81%) and toluene (6.76%) were the characteristic emission components of the furniture manufacturing industry in this study. Similar to the furniture manufacturing industry, propane (13.98%), toluene (9.08%) and ethanol (5.87%) were also the characteristic emission components of the vehicle manufacturing industry. In addition, butenone (9.18%) also constituted a significant proportion in the vehicle manufacturing industry.

### 3.3. Reactivity of VOC Species

The calculation results of the ozone generation potential (OFP) are shown in Figure 4 below. The OFP values in the emission links of various industries ranged from 0.75 to 157.51 mg·m^−3^. The maximum OFP appeared in the packaging and printing industry, mainly contributed by highly active OVOCs, indicating that the packaging and printing industry has a higher potential for ozone generation. Overall, the packaging and printing industry has a relatively higher reactivity among the five industries in this study. The OFP value was not positively correlated with the emission concentration. In the link with a higher OFP value, alkenes, aromatics and OVOCs have obvious emissions. The plastic products industry had the highest OFP value in the production workshop, mainly contributed by acetone, 1-butene and pentane. The packaging and printing industry mainly contributed to OVOCs (80%) before the treatment of sewage chimneys, while other process links mainly contributed to olefins and aromatic hydrocarbons. The OVOC components in the ink industry had high emissions, but because the main emission component was ethyl acetate and the MIR value of ethyl acetate was small, the overall OFP value of the ink industry was not high, the contribution of aromatic hydrocarbons in the finished product storage was 74%, and OVOCs between grinding was 50%. The overall OFP value of the furniture manufacturing industry was relatively low, and each process had no significant contribution. The OFP value in the vehicle manufacturing industry was the highest in the painting process, mainly contributed by highly active aromatic hydrocarbons. Other process segments were mainly contributed by aromatic hydrocarbons and alkenes. It can be seen that OFP is mainly determined by the MIR value of the species and secondarily affected by the emission concentration of VOCs.

Parameters, such as SR and SOA, have been widely used to evaluate the impact of VOC emissions on the formation of secondary pollutants [37,38]. In this article, the focus is on sources of the top 10 compounds that contribute to SR and SOA. Figure 5 below shows the top 10 VOC components contributed by SR and SOA. The species contained in SR were alkanes, alkenes, aromatics and OVOCs, while SOAs were alkanes and aromatics. The VOC with the most photochemical formation trend for ozone was toluene, which came from plastic products (12%), packaging and printing (19%), ink (24%), furniture manufacturing (19%) and vehicle manufacturing (26%). m-Parylene, propionaldehyde and o-xylene contributed significantly to the plastic products industry, with contribution rates of 48, 47 and 55%, respectively. The components that contributed the most to the packaging and printing industry were propionaldehyde, 1-pentene and n-hexanal, with contribution rates of 44, 85 and 84%, respectively. Almost all ethyl acetate came from the ink and furniture manufacturing industries. Overall, the packaging and printing, and ink-use industries were the two sectors that contributed the most to SR among the top 10 VOC components.

The highest component in SOA formation trends was toluene, which came from the plastics (10%), packaging and printing (16%), inks (37%), furniture manufacturing (16%) and vehicle manufacturing industries (21%). The contribution rates of m-parylene, ethylbenzene and o-xylene to the plastic products industry were 47, 38 and 56%, respectively, and at the same time, they made a significant contributions to the vehicle manufacturing industry, with contribution rates of 26, 35 and 26%, respectively. Except for toluene, m/p-xylene, ethylbenzene and o-xylene, the other components had no obvious tendency to form SOAs.

Among the VOC components with larger SRs and SOAs, toluene, o-xylene and m/p-xylene all had significant contributions to SR and SOA, indicating that toluene, o-xylene and m/p-xylene made significant contributions to SR or SOAs. The main contribution components of SR and SOA in the top 10 emissions were significantly different, indicating that the VOC components that significantly contributed to the formation of SR may not contribute greatly to the formation of SOAs at the same time. Therefore, VOCs and industries regulated using emission-based control strategies will be very different from SR and SOA-based strategies.

### 3.4. Health Risk Assessment

Human health risk assessment is used to estimate the adverse health effects of human exposure to pollutants. Among the VOC species monitored in this study, the species shown in Appendix A belong to the toxic and harmful air pollutants announced by the United States Environmental Protection Agency (US EPA). Enterprise workers and surrounding residents are mostly exposed to the factory area, workshop and ambient air. Therefore, the US EPA health risk assessment model was used to evaluate the health risk of fugitive emissions exposure groups. In the US EPA risk assessment, the greater the hazard index (HI) value is, the greater the risk. When the hazard index (HI) > 1, chronic health hazards occur after long-term exposure. The VOC components included in the EPA risk assessment are mainly concentrated on alkanes and aromatic hydrocarbons, and the risk assessment cannot be comprehensively carried out. There are potential uncertainties, but harmful components, such as BTEX, have been covered. According to the organic pollutants with non-carcinogenic risk listed by the EPA, combined with the VOC species detected by the company, 22 organic pollutants that may cause non-carcinogenic risk were finally screened (Appendix A). The total HI values of plastic products, packaging printing and ink enterprises were relatively high, 2.45, 2.53 and 1.59, respectively, all of which exceed the EPA specified limit (see Figure 6 below). The maximum HI value appeared in the indoor production workshop and raw material storage area of the packaging and printing industry. This shows that the VOCs emitted by the company pose non-carcinogenic risks to the human body. The HI values of the furniture manufacturing and vehicle manufacturing enterprises were 0.11 and 0.04, respectively, indicating that the unorganized escape of VOCs in these industries is unlikely to cause chronic health risks with no significant non-carcinogenic risk. Whereas pentane, heptane, nonane, 1,3-butadiene, 1,2-dichloropropane, 1,1,2-trichloroethane, benzene, toluene, m-para-xylene and o-xylene contribute to the top 10 VOC components, accounting for more than 88% of the industry’s HI value. Therefore, from the perspective of human health, it is necessary to adopt control strategies for VOC components in workshops of plastic products, packaging printing and ink enterprises, such as strengthening personal protection, improving ventilation systems, increasing indoor air exchange rates [39,40], and reducing the impact of VOC components in workshops on human health.

## 4. Conclusions

(1)This study quantified the concentrations of VOC emissions from the plastic products, packaging and printing, printing ink, furniture manufacturing, and vehicle manufacturing industries, The results showed that the concentrations of VOC emissions were 3.48–50.91 mg·m^−3^ for packaging and printing, 2.88–43.49 mg·m^−3^ for printing ink, 0.88~12.35 mg·m^−3^ for automobile manufacturing, 2.97~11.38 mg·m^−3^ for plastic products and 1.08~2.25 mg·m^−3^ furniture manufacturing industries. Overall, alkanes, aromatics and OVOCs were the most important emission sources.(2)Based on the field sampling analysis of VOCs, we provided information of VOC emission components, generation potential and hazard index in Yuncheng city. The effects on human health and on ozone and SOA formation were quantitatively assessed by calculating the values of VOC components OFP, SR, SOA and HI. From the analysis of different results, aromatics had different degrees of negative effects on the atmospheric environment and human health. There are significant differences between the top 10 OFPs and the main contributing components of SOAs. However, toluene, o-xylene and m/p-xylene had a significant tendency to form OFP or SOAs. In addition, 1-pentene, propanal, and ethyl acetate also had a great tendency to form ozone. Considering the potential harm of VOC emissions on the human body, a health risk assessment was also carried out. Pentane, 1,3-butadiene, 1,2-trichloroethane, benzene, p-xylene and o-xylene had large risk values, and continuous and long-term monitoring of VOCs emission characteristics in the workshop is needed to fully reveal the health risks of future work.(3)In order to establish complete and accurate VOC emission characteristics, it is necessary to carry out a large number of studies to overcome the obstacles of the complex and diverse chemical compositions in VOC research, and screen for priority control pollutants in order to further control VOC emissions, accurately identify characteristic pollutants of various emission sources, and provide valuable information for VOC control and emission reduction policies.

## Figures and Tables

**Figure 1 ijerph-20-03517-f001:**
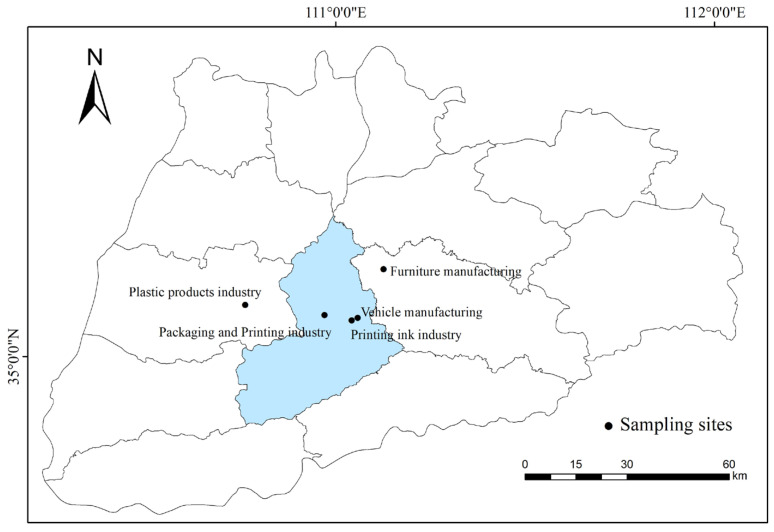
Sampling point area bitmap.

**Figure 2 ijerph-20-03517-f002:**
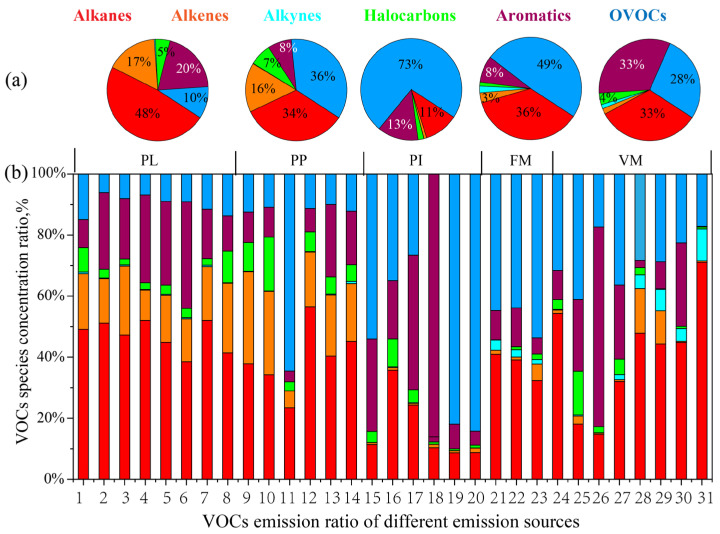
Composition characteristics of VOCs emitted by industry (**a**) and different process links (**b**). (Note: PL: plastic products industry; PP: packaging and printing industry; PI: printing ink industry; FM: furniture manufacturing; VM: vehicle manufacturing industry).

**Figure 3 ijerph-20-03517-f003:**
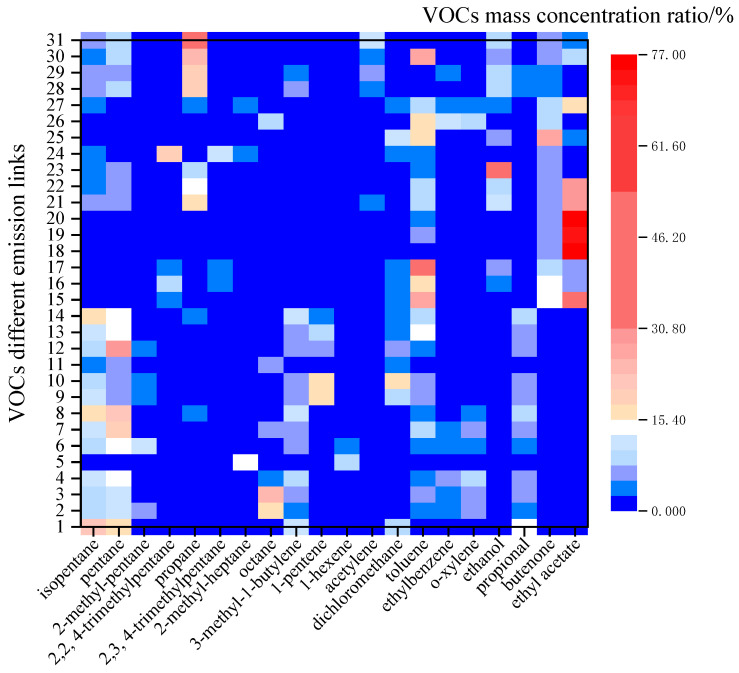
Main species of process links in the different industries.

**Figure 4 ijerph-20-03517-f004:**
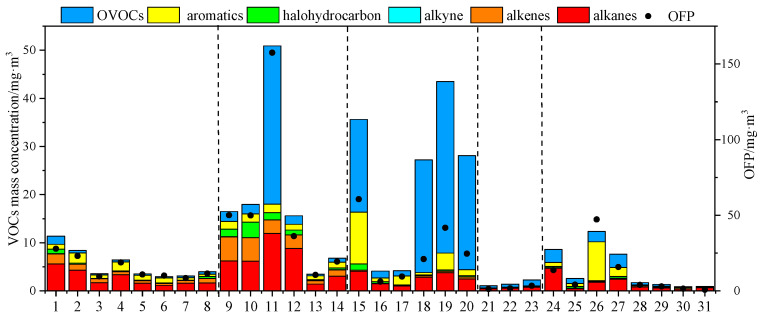
VOC emission concentrations and OFPs in various industries.

**Figure 5 ijerph-20-03517-f005:**
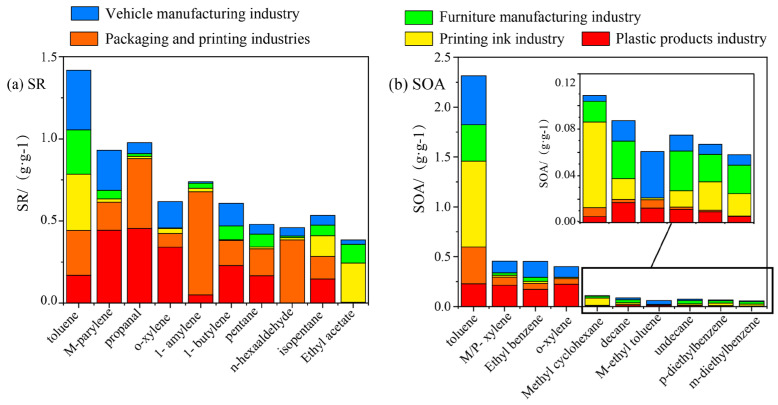
Component contributions of the top 10 VOCs in SR (**a**) and SOA (**b**) in the different industries.

**Figure 6 ijerph-20-03517-f006:**
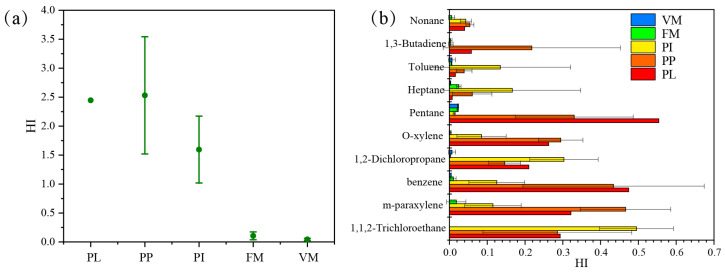
HI (**a**) and VOC (**b**) components of priority control in different industries.

**Table 1 ijerph-20-03517-t001:** Detailed information and emission concentrations of VOC industry emission sources.

Industry Type	Numbering	Sampling Link	Pollution Type	Processing Technology
Plastic products industry	1	Drawing plant	Escape	
	2	Before the wire drawing returns to the exhaust port	Organized emission	Plasma + activated carbon adsorption
	3	After the wire drawing returns to the exhaust port		
	4	Print-slitting machine exhaust port before treatment		
	5	Print-slitting machine exhaust port after treatment		
	6	Blow molding before the exhaust port		
	7	After blowing the exhaust port		
	8	The exhaust port of inner adhesive outer coating machine		
Packaging and Printing industry	9	Production workshop (indoor)	Escape	
	10	Production workshop (raw material area)		
	11	Before discharge chimney treatment	Organized emission	Photooxygen plasma + activated carbon adsorption
	12	After the sewage chimney treatment		
	13	Raw materials storage room	Escape	
	14	Hazardous waste storage room		
Printing ink industry	15	Finished product storage room	Escape	
	16	Raw materials storage room		
	17	Hazardous waste storage room		
	18	Ingredient distribution room		
	19	Workshop		
	20	Filter packing room		
Furniture manufacturing	21	Glue workshop	Escape	
	22	Coating workshop		
	23	Drain the chimney	Organized emission	
Vehicle manufacturing	24	Electrophoresis drying	Organized emission	Combustion engine
	25	Plastic parts spray painting		Zeolite runner concentration + RTO
	26	The framed painting		Combustion engine
	27	Touch-up room		Activated carbon adsorption
	28	Spray paint—electrophoresis drying		Combustion engine
	29	Medium coat—top coat drying		
	30	Outside the painting shop	Escape	
	31	Outside the paint workshop		

## Data Availability

The datasets used or analyzed during the current study are available from the corresponding author upon reasonable request.

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
