# Peer review of "Study on the Fingerprint and Atmospheric Activity of Volatile Organic Compounds from Typical Industrial Emissions"

_ijerph, 2023, doi:10.3390/ijerph20043517_

Round 1

Reviewer 1 Report

This manuscript reports the fingerprint, atmospheric activity and health risk of VOCs from five different industrial emissions, including plastic products industry, packaging and printing industry, printing ink industry, furniture manufacturing industry and vehicle manufacturing industry. These results provided important information for environmental pollution issues. One of the critical concerns was that if the characteristics of these five kinds of industrial emissions were typical. The authors mentioned the source spectrum of VOC emissions from industries has been studied in many cities in China, were the results from this study consistent with previous works? Another bias was that the method used to assess the carcinogenic risk of VOCs was not definite, the cancer slope or some statistics about the morbidity of these VOCs were not mentioned.

Specific comments:

1.     Figure 1,the industry type of each site should be given in this map. And, this map could not illustrate “The research industry selected in this study was located in the central area of Yuncheng city, which is a relatively densely industrialized area.”

2.     In the methodology section, the dynamic retention efficiency of the sampling material used in this work should be given. 

3.     Only one sample was collected in each numbered site, how to ensure the repeatability and representative each sample? 

4.     The mass concentration or mixing ratios of each VOC detected in the work should be provide. 

5.     In Figure 2, VOC compositions in orange and red are all alkanes? Furthermore, yellow is not a clear color.

6.     Figure 1, Figure 3 were not mentioned in the main text. Figure 3 needs more explanation.

7.     The small figure in Figure 5(b) was used to illustrate SR or SOAP?

Reviewer 2 Report

Please find the review comments attached.

Round 2

Reviewer 1 Report

The authors have almost successfully revised the paper based on the suggestions. But the resolution of some figures are still too low, such as Figure 2 and Figure 5, the values on these figures were too fuzzy.

Author Response

The fuzzy numbers in Figures 2 and 5 have been corrected.

Reviewer 2 Report

Authors have revised the manuscript successfully. t can be accepted for the publication

Author Response

Thank you for your suggestions on this paper.